# Transcriptional CDK Inhibitors as Potential Treatment Option for Testicular Germ Cell Tumors

**DOI:** 10.3390/cancers14071690

**Published:** 2022-03-26

**Authors:** Kai Funke, Robert Düster, Prince De-Graft Wilson, Lena Arévalo, Matthias Geyer, Hubert Schorle

**Affiliations:** 1Department of Developmental Pathology, Institute of Pathology, University Hospital Bonn, 53127 Bonn, Germany; kai.funke@ukbonn.de (K.F.); sharkalemagh@gmail.com (P.D.-G.W.); lena.arevalo@ukbonn.de (L.A.); 2The Institute of Structural Biology, University of Bonn, 53127 Bonn, Germany; robert.duster@mssm.edu (R.D.); matthias.geyer@uni-bonn.de (M.G.)

**Keywords:** testicular germ cell tumors, transcriptional cyclin-dependent kinase inhibitors, CDK inhibitors, NVP2, SY0351, YKL-5-124

## Abstract

**Simple Summary:**

Type II testicular germ cell tumors are a severe type of cancer in young men demanding alternative treatment options to conventional chemotherapy with less side effects. In particular, patients with chemotherapy-resistant tumors face a bad prognosis and low survival rates. In other tumor entities, transcriptional cyclin-dependent kinases (7/8/9/12/13) have been demonstrated to be effective targets. Here, we studied the effects of transcriptional cyclin-dependent kinase inhibitors on a cellular and molecular level. We found several inhibitors to be highly cytotoxic for certain testicular germ cell tumor cell lines while leaving a somatic (fibroblast) control cell line unaffected. This opens up a novel field for effective and specified treatment of type II testicular germ cell tumors.

**Abstract:**

Type II testicular germ cell tumors (TGCT) are the most frequently diagnosed solid malignancy in young men. Up to 15% of patients with metastatic non-seminomas show cisplatin resistance and a very poor survival rate due to lacking treatment options. Transcriptional cyclin-dependent kinases (CDK) have been shown to be effective targets in the treatment of different types of cancer. Here, we investigated the effects of the CDK inhibitors dinaciclib, flavopiridol, YKL-5-124, THZ1, NVP2, SY0351 and THZ531. An XTT viability assay revealed a strong cytotoxic impact of CDK7/12/13 inhibitor SY0351 and CDK9 inhibitor NVP2 on the TGCT wild-type cell lines (2102EP, NCCIT, TCam2) and the cisplatin-resistant cell lines (2102EP-R, NCCIT-R). The CDK7 inhibitor YKL-5-124 showed a strong impact on 2102EP, 2102EP-R, NCCIT and NCCIT-R cell lines, leaving the MPAF control cell line mostly unaffected. FACS-based analysis revealed mild effects on the cell cycle of 2102EP and TCam2 cells after SY0351, YKL-5-124 or NVP2 treatment. Molecular analysis showed a cell-line-specific response for SY0351 and NVP2 inhibition while YKL-5-124 induced similar molecular changes in 2102EP, TCam2 and MPAF cells. Thus, after TGCT subtype determination, CDK inhibitors might be a potential alternative for optimized and individualized therapy independent of chemotherapy sensitivity.

## 1. Introduction

Type II testicular germ cell tumors (TGCT) are the most prevalent malignancies in young men aged 18 to 35 years [1] with rising incidence in Western countries in particular [2]. TGCTs are classified into seminomas and non-seminomas, which are further subdivided into embryonal carcinomas (EC), yolk sac tumors (YST), teratomas (Ter) and choriocarcinomas (Cc) [3,4,5]. ECs represent the totipotent stem cell population of these tumors able to further differentiate into YST, Ter and Cc. [5,6]. Both sub-entities, seminomas and non-seminomas, arise from primordial germ cells (PGCs) which are developmentally arrested and eventually turn into the precursor lesion termed germ cell neoplasia in situ (GCNIS) [5,7,8]. Seminomas closely resemble PGCs and GCNIS in histology, global gene expression and epigenetic pattern. Due to limited differentiation abilities, seminomas are considered as the default pathway of defect PGCs [9]. Although high curation rates of more than 90% are achieved by orchiectomy, followed by platinum-based chemo- or radiotherapy, 10–15% of patients with metastatic TGCTs fail multiple-line treatment due to drug resistance, ending up with poor prognoses and short survival of only a few months [2,10,11]. Therefore, further treatment options for TGCTs are needed which do not only have a good prognosis for refractory tumor patients but also as an alternative for cisplatin-based chemotherapy aiming for reduced side effects.

Cyclin-dependent kinases (CDKs) are a promising target for cancer treatment. Cell-cycle-associated kinases (CDK1/2/4/6) directly regulating cell cycle progression have already been extensively studied in preclinical and clinical trials using different CDK inhibitors in TGCTs and other types of cancer [12,13,14]. In contrast, data regarding inhibitors of the transcriptional-associated CDKs (CDK7/8/9/12/13, tCDKs) are scarce; hence, their use as therapeutic targets is not that advanced [13]. CDKs7/8/9/12 and 13 are key factors for RNA polymerase II (RNA Pol II) activity. CDK7, its partner cyclin H and the accessory protein MAT1 form the CDK-activating kinase (CAK) complex. CAK-mediated phosphorylation of the carboxy-terminal domain (CTD) at specific serine residues of the DNA-directed subunit of RNA Pol II initiates RNA-Pol-II-dependent transcription. Thereby, promotor escape is initiated, which is the transition of RNA Pol II from promotor binding to advanced downstream regions of template DNA [15,16]. The post-initiation process of RNA Pol II progression is immediately interrupted, causing a pause state. The kinase activity of CDK9 is activated by binding to cyclin T, forming the positive transcription elongation factor b (P-TEFb), which selectively phosphorylates the CTD of RNA Pol II and the negative elongation factor (NELF), promoting the release of RNA Pol II pausing, resulting in transcript elongation [17,18]. CDK12 and CDK13 both partner with cyclin K, thereafter establishing a specific phosphorylation pattern of the CTD of RNA-Pol II, revealing progressed transcription elongation of full-length mRNA by prevention of intronic polyadenylation [19,20].

Flavopiridol, a pan-CDK inhibitor (CDK1/2/4/6/7/9), has been analyzed in various types of tumors [14]. Apoptosis induction by flavopiridol was shown in non-seminoma cell lines (NT2, 2102EP, NCCIT) [21]. Additionally, a clinical phase I study was performed with patients suffering from refractory germ cell tumors, revealing a partial and highly patient-specific response to flavopiridol application [22]. The pan-CDK inhibitor dinaciclib (CDK1/2/5/9/12/13) induced apoptosis in ovarian cancer cell lines and synergizes with cisplatin [23]. Of note, a clinical phase II study investigated the potential of dinaciclib for treating advanced breast cancer, which indicated good tolerability but displayed no advantage over standard therapy [24].

Recent advances resulted in a toolbox of highly potent and selective inhibitors towards transcriptional CDKs, allowing for the analysis of their contribution to transcriptional regulation and, at the same time, opening new avenues for cancer treatment. In this regard, covalent inhibition of tCDKs7/12/13 by THZ1, initially launched as a CDK7 selective covalent inhibitor, has been a major breakthrough since it potently induces apoptosis across a wide variety of cancer cell lines [25,26]. Based on THZ1, a highly selective CDK12/13 inhibitor, THZ531, was derived, which had strong apoptotic effects in Jurkat cells [27]. Due to CDK12’s role in the expression of DNA damage repair genes, CDK12 inhibition is able to induce a “BRCAness” phenotype and hence can act synergistically with DNA-damaging agents [28]. SY0351, a THZ1-based inhibitor with improved selectivity towards CDK7 showed efficient antitumor effects in multiple AML xenograft models [29] and extended previous views of CDK7 as a direct activator for the CDK-activating kinase for CDKs9/12 and 13 [30].

Due to the lack of selectivity of THZ1-based compounds for CDK7 targeting, the laboratory of Nathanael Gray combined a PAK4 inhibitor scaffold with the covalent THZ1 linker, resulting in a highly selective CDK7 inhibitor without cross-reactivity to CDK12/13. CDK7 inhibition by YKL-5-124 alone is not able to induce apoptosis in HAP1 but does so when combined with CDK12/13 inhibition [31]. Moreover, a study using YKL-5-124 in small-cell lung cancer cells induced cell cycle arrest and genomic instability, triggering antitumor immunity [32].

NVP2 is a selective ATP-competitive CDK9 inhibitor [33]. CDK9 inhibition was proven to be a powerful anticancer tool, as shown for different malignancies (ovarian cancer [34], prostate cancer [35], human head and neck squamous cell carcinoma [36] and gastric cancer [37]). The proteolysis-targeting chimera (PROTAC) THAL-SNS-032 selectively degrades CDK9. Interestingly, the CDK9 inhibitor and CDK9 degrader have been compared using MOLT4, a cell line derived from acute lymphoblastic leukemia, inducing apoptosis upon administration of both compounds. However, THAL-SNS-032 PROTAC leads to delayed but prolonged cellular response compared to NVP2 [33].

In this study, we analyzed the effects of flavopiridol, dinaciclib, THZ1, YKL-5-124, SY0351, THZ531, NVP2 and THAL-SNS-032 on different TGCT cell lines to identify and characterize novel specific treatment opportunities for TGCTs, aiming for reduced side effects compared to platinum-based treatment and to open up a new therapeutic option for patients with cisplatin-resistant or refractory TGCTs. We found NVP2 and SY0351 to be most effective at inducing apoptosis in seminomas and EC cells applying low nanomolar concentrations, suggesting that inhibition of CDK9 and CDK7/12/13 might be an alternative treatment option for TCGT.

## 2. Materials and Methods

### 2.1. Cell Culture

The following cell lines were used 2102EP, 2102EP-R, NCCIT, NCCIT-R (all ECs), TCam2 (seminoma cell line), FS1 (Sertoli cell line) and MPAF (human adult fibroblast cell line). Cells were cultivated as described previously [38,39]. In brief, the ECs were cultivated in Dulbecco’s modified Eagle’s medium (DMEM) supplemented with 10% fetal bovine serum (FBS), 1% penicillin–streptomycin and 200 mM L-glutamine. FS1 cells were grown in DMEM supplemented with 20% FSB, 1% penicillin–streptomycin, 200 mM L-glutamine and 1% nonessential amino acids (NEAA). MPAF cells were cultured in DMEM supplemented with 10% FBS, 1% penicillin–streptomycin, 200 mM L-glutamine and 1% NEAA. TCam2 cell line was cultured in RPMI medium supplemented with 10% FBS, 1% penicillin–streptomycin and 200 mM L-glutamine. All cell lines were kept at 37 °C and 7.5% CO_2_. 2102EP and NCCIT were a kind gift from Prof. Dr. L. Looijenga (Princess Máxima Center for Pediatric Oncology, Utrecht, The Netherlands). Cisplatin-resistant lines 2102EP-R and NCCIT-R were obtained from PD Dr. F. Honecker (Breast and Tumor Center, ZeTup Silberturm, St. Gallen, Switzerland). MPAFs were a kind gift from PD Dr. M. Peitz (Bonn University, Institute of Reconstructive Neurobiology, Bonn, Germany). FS1 cells were obtained from Prof. Dr. V. Schumacher (Boston Children’s Hospital, Department of Urology; Harvard Medical School, Department of Surgery, Boston, MA, USA).

### 2.2. Cell Viability Assay

The assay was performed based on the reduction in XTT tetrazolium salt as described previously [40]. In summary, 3000 cells were seeded per well of a 96-well cell culture plate in 100 µL of cell culture medium. The next day, cells were treated with CDK inhibitor or CDK degrader for 24/48/72 h. For XTT assay, 0.6 mg/mL XTT sodium salt was dissolved in cell culture medium (DMEM or RPMI) and supplemented with 25 µM electron carrier N-methyl dibenzopyrazine methyl sulfate (PMS). A total of 50 µL of XTT/PMS was added to each well and incubated for 4 h at 37 °C, 7.5% CO_2_. For readout, absorption was measured at 450 nm and 650 nm. Specific XTT signal (450 nm) was subtracted by nonspecific absorbance (650) and blank (cell culture medium). Averages of technical triplicates were calculated, and treated sample was referenced to the corresponding dimethyl sufoxide (DMSO) control. Experiment was repeated several times to obtain biological replicates. For statistical analysis, two-tailed Student’s *t*-test was applied. Datapoints with *p*-value < 0.05 indicated significantly changed viability of treated compared to control group and labeled with an asterisk.

### 2.3. Protein Extraction, SDS-PAGE and Western Blot

Protein extraction was performed on untreated cells with RIPA buffer (Cell signaling, Danvers, MA, USA) supplemented with cOmplete ULTRA Tablets protease inhibitor (Roche, Swiss). After sonication, the lysate was centrifuged at 13,000× *g* at 4 °C for 15 min. Protein concentration in the clear supernatant was determined by Pierce BCA protein assay kit (Thermo Fisher Scientific, Waltham, MA, USA) according to manufacturer’s manual. SDS-polyacrylamide gel electrophoresis (SDS-PAGE) and Western blot analysis were performed as described previously [39]. In summary, after separation of denatured proteins via SDS-PAGE according to their molecular weight, proteins were transferred onto a Roti PVDF membrane with pore size of 0.45 µm (Carl Roth, Karlsruhe, Germany) using Trans-Blot^®^ Turbo™ Transfer System (Bio-Rad Laboratories, Hercules, CA, USA) for 30 min at 2.5 ampere and 25 volts. Target-specific primary and species-specific HRP-linked secondary antibodies were diluted according to Table 1. Detection was performed using the ECL substrate WESTAR NOVA 2.0 (Cyanagen, Bologna, Italy) in combination with the ChemiDoc MP imaging system (Bio-Rad Laboratories, Hercules, CA, USA). β-Actin was used as the loading control.

### 2.4. 7AAD/AnnexinV (Apoptosis) and Hoechst-FACS Analysis (Cell Cycle)

Cells were seeded in 6-well cell culture plates (1.5 × 10^5^ cells/well). The next day, cells were treated with a CDK inhibitor (NVP2 and SY0351 10 nM, YKL-5-124 and THZ531 100 nM) or DMSO (0.0002%). For cell cycle analysis, cells were harvested after 20 h of treatment and resuspended in PBS. Ice-cold methanol was added dropwise while shaking until 70% methanol *v*/*v* was reached. Cells were incubated for 2 h at 4 °C and subsequently washed twice with cold PBS. Staining (2 µg/mL Hoechst-33342, 50 µg/mL RNaseA in PBS) was performed for 30 min at 37 °C. The cell cycle distribution was measured in an FACS Canto (BD BioSciences, Heidelberg, Germany) and analyzed using FlowJo™ v10.8 Software (BD BioSciences, Heidelberg, Germany). For apoptosis analysis, PE AnnexinV Apoptosis Detection Kit with 7-AAD (BioLegend, San Diego, CA USA) was used according to the manufacturer’s manual. Measurement was performed in an FACS Canto while analysis was performed in the corresponding BD FACSDiva software^TM^ (BD BioSciences, Heidelberg, Germany). Two-tailed Student’s *t*-test was applied for calculation of significance.

### 2.5. RNA Sequencing Analysis

Cells were seeded into 6-well cell culture plates (2 × 10^5^ cells/well). Treatment was started the next day for 1 h or 24 h with 10 nM NVP2, 10 nM SY0351, 100 nM YKL-5-124, 100 nM THZ531 or DMSO. RNA was isolated using the RNeasy Mini Kit (Qiagen, Hilden, Germany). To determine RNA integrity (RIN), Nano 6000 Assay kit with the Agilent Bioanalyzer 2100 system (Agilent Technologies, Santa Clara, CA, USA) was used. Only samples with RIN > 7 were used for RNA sequencing analysis. RNA quality control, library preparation (QuantSeq 3′-mRNA Library Prep (Lexogen, Vienna, Austria)) and RNA sequencing were performed by the Core Facility Next Generation Sequencing (University of Bonn). An Illumina HiSeq 2500 V4 device (producing >10 million, 50 bp 3′-end reads per sample) was used. Raw data quality was initially investigated using FastQC [41]. Further, sequences were trimmed using TrimGalore [42]. Trimmed sequences were mapped to the human genome (GRCh38.p13) using HISAT2.1 [43]. Quantification and annotation of transcripts was performed by using StringTie 1.3.3 application [44]. To prepare a DEseq2 compatible data table, the Python script preDE.py included in the StringTie package was used. Further analyses were conducted in R/Bioconductor [45,46] embedded in R-studio environment [47]. The DESeq2 1.16.1 package [48] was used for calculation of differentially expressed genes with an adjusted *p* value < 0.05 (Benjamini–Hochberg method). The volcano plots were prepared based on the differential expression data using ggplot2 3.3.3 [49]. Differential expression data were further analyzed using STRING 11.5 database [50], integrated Gene Ontology [51,52], Reactome [53] and KEGG pathway [54] analysis tools. Venn diagrams were assembled using Venny 2.1 software [55].

### 2.6. Peptide Chip Array

The protein extraction and kinase activity assay were performed as described elsewhere [56]. In summary, the cells were seeded and treated as described for RNA sequencing analysis. Protein extraction was performed with M-PER Mammalian Protein Extraction Reagent, supplemented with Halt™ Phosphatase Inhibitor Cocktail and Halt™ Protease-Inhibitor-Cocktail (all Thermo Fisher Scientific, Waltham, MA, USA). Samples were analyzed in biological duplicates. Protein samples were investigated for serine and threonine kinase activity using a Pamstation provided by Pamgene (‘s-Hertogenbosch, The Netherlands). Lysates were diluted in assay buffer containing 400 µM ATP and incubated on an immobilized peptide array. Phosphorylated peptides were detected in a two-step process comprising primary antibodies directed against phosphorylated serine and threonine residues, followed by detection with an FITC-labeled secondary antibody for fluorescent readout. Signal intensities were monitored by analysis of the fluorescent signal. Analysis of upstream kinases is based on the peptide array phosphorylation pattern and was carried out by Pamgene application specialists using the Bionavigator software (Pamgene).

## 3. Results

### 3.1. Cell Cycle and Transcriptional CDKs Are Expressed in TGCT Cell Lines

To investigate the expression of CDKs in testicular germ cell tumors, we performed meta-analyses of microarray data on different cell lines and tissues previously published by us [40,57]. All TGCT cell lines analyzed (TCam2, seminomas; 2102EP, NCCIT both embryonal carcinomas (EC); JAR, choriocarcinomas) and control cell lines (Sertoli cell line FS1, fibroblast cell line MPAF) showed moderate to high expression of CDKs1/2/4/5/6/7/9/10/11A/11B/12 and 13 (Figure 1A). In testicular germ cell tumor tissues (germ cell neoplasia in situ, seminomas, ECs, teratomas, mixed non-seminomas) and normal testis tissue, a high expression of CDK1/4/7 and 9 was observed (Figure 1B). To confirm the expression of CDKs7/9/10/12 and 13 on protein levels, Western blot analyses were performed (Figure 1C). Interestingly, the protein levels varied considerably between the cell lines. While 2102EP, NCCIT, TCam2 and FS1 revealed medium to high levels of CDKs tested, MPAF cells showed only very low CDKs9/10/12 and 13 protein levels. A high level of CDK7 protein was detected in all cell lines. Double bands represent different CDK7 phospho-isoforms.

### 3.2. CDK Inhibitors Display Cytotoxic Effect on TGCT Cell Lines

Next, we analyzed the effect of seven CDK inhibitors (NVP2, SY0351, YKL-5-124, THZ531, THZ1, flavopiridol, dinaciclib) and one CDK degrader (THAL-SNS-032) on the viability of three different parental TGCT cell lines (2102EP, NCCIT, TCam2), two cisplatin-resistant subclone lines (2102EP-R, NCCIT-R) and two control cell lines (FS1, MPAF) via an XTT assay (Figure 2, Appendix A). The CDK9 inhibitor NVP2 effectively reduced the viability already at a concentration of 10 nM in the TGCT cell lines and in the FS1 cell line. Higher concentrations led to an even stronger reduction in viability. Cisplatin-resistant cells are affected in a similar range as the parental cell lines. The CDK9 degrader THAL-SNS-032 showed comparable reduction in viability at concentrations of 100 nM and 500 nM. Of note, the MPAF and FS1 cells were less affected by CDK9 degradation compared to CDK9 inhibition. THAL-SNS-032 revealed a reduction in viability for FS1 cells only at 500 nM. MPAF cells were not affected at 1 and 10 nM and showed a minimal decrease in viability at 100 and 500 nM.

The EC cell lines (2102EP, 2102EP-R, NCCIT, NCCIT-R) seemed to be highly sensitive to dinaciclib and flavopiridol, indicated by a strong decrease in viability at concentrations of 1 and 50 nM, respectively. While TCam2 and FS1 cells showed only a moderate sensitivity, MPAF cells were almost resistant to these drugs.

Interestingly, the CDK7 inhibitor YKL-5-124 reduced cell viability in 2102EP cells selectively while all other cell lines showed lower sensitivity towards this compound. MPAF cells were especially unaffected by YKL-5-124 even at the highest concentration tested (500 nM). THZ1 and SY0351, both inhibiting CDK7, led to decreased viability while SY0351 caused higher cytotoxicity at only 10 nM compared to THZ1. THZ531 was only effective at high concentrations (100 and 500 nM).

To further analyze these findings, the IC50 values were calculated for each time point, each cell line and each inhibitor (Appendix A, Table 2). Supporting the initial analysis, a time dependency was confirmed by the IC50 values.

The lowest IC50 values were observed for dinaciclib treatment of tumor cell lines at 72 h between 0.8 and 4.3 nM, indicating a very high potency while TGCT cell lines appeared to be less sensitive towards flavopiridol (IC50 at 72 h between 7.5 and 273 nM). Based on these data, SY0351 (IC50 at 72 h 6.4 to 11.6 nM) has a much stronger effect on the tumor cells compared to THZ1 (IC50 at 72 h 6.5 to 847 nM). IC50 values for YKL-5-124 are low for 2102EP/2102EP-R cells (72 h 17.2 nM 2102EP and 23.6 nM 2102EP-R) and clearly higher for all other cell lines. NVP2 is also considered highly effective with IC50 values ranging from 6.1 to 16.1 nM for the tumor cell lines (72 h). The CDK9 degrader THAL-SNS-032 showed higher IC50 values (30 to 40 nM, 72 h in TGCT cells). The highest IC50 values were observed for THZ531 (from 74.5 to greater than 1000 nM at 72 h for tumor cell lines). Of note, the fibroblast control cell line MPAF tolerated high levels of these CDK-inhibitors, showing IC50 values above 1000 nM independent of the inhibitor and the time. These results indicate that a treatment using CDK inhibitors might specifically impinge on germ cell tumors, leaving somatic cells unaffected.

### 3.3. NVP2, SY0351, YKL-5-124 and THZ531 Affect Cell Cycle Progression and Induce Apoptosis in TGCT Cells

For investigation of the cellular and molecular effects, we decided to focus on the four inhibitors NVP2, SY0351, YKL-5-124 and THZ531. We determined changes in cell cycle progression by Hoechst staining using fluorescence-activated cell sorting (FACS) after 20 h of inhibitor treatment (Figure 3A, Appendix A). In general, NVP2, SY0351, YKL-5-124 and THZ531 displayed only mild effects on the cell cycle after 20 h of inhibitor treatment.

Exposure to NVP2 resulted in elevated G1-phase populations in the EC cell lines (2102EP, NCCIT) and increased the fraction of cells in the G1 and G2/M phases for the TCam2 cell line with a reduced number of cells in the S phase, accordingly. In the control cell line FS1, only a small fraction of cells was shifted to the G1 and G2/M phases. Similar effects were observed after SY0351 treatment, i.e., reduction in cells in the S phase, accumulation of cells in the G1 phase in the EC cell lines and G1- and G2/M-phase enrichment of TCam2 cells. The cell cycle of FS1 cells was hardly affected while MPAF cells showed reduced fraction of cells in the G1 phase. Treatment with YKL-5-124 resulted in reduction in 2102EP and TCam2 cells in the S phase but induced a strong increase in the S phase for NCCIT. Further, slightly elevated number of NCCIT cells were found in the G2/M phase, indicating a G2/M-phase cell cycle arrest with an overrepresentation of cells in the S phase which had not reached the G2/M phase after 20 h of treatment. For 2102EP and TCam2 cells, an elevated number of cells were observed in the G2/M phase and G1 phase, respectively. Of note, data from FS1 and MPAFs suggest cell cycle progression. THZ531 led to a highly increased number of cells in the G2/M phase for the 2102EP and NCCIT cell lines. TCam2 and MPAF cells showed hardly any changes in terms of cell cycle distribution, which might originate from their longer doubling times [57].

Next, we measured the effect of NVP2, SY0351, YKL-5-124 and THZ531 on apoptosis by 7AAD/AnnexinV FACS analysis after 24 and 48 h inhibitor treatment (Figure 3B, Appendix A). NVP2 treatment of 2102EP and NCCIT cells increased apoptosis up to 5.5-fold compared tothe control. In TCam2, FS1 and MPAF cells, the fold change in apoptotic cells (up to 3.3-fold) was lower. SY0351 treatment revealed similar results. The strongest apoptosis induction was seen in the 2102EP and NCCIT cells, while less apoptosis was observed in the other tested cell lines (TCam2, MPAF, FS1). YKL-5-124 effects were less pronounced (up to 2.8-fold change over DMSO control) compared to NVP2 and SY0351. Interestingly, FS1 and MPAF cells showed almost no difference between treatment and control, indicating that YKL-5-124 exclusively targets tumor cell lines. THZ531 led to a moderate increase in apoptosis induction in tumor cell lines. A strong increase in apoptotic cells compared to the control was observable only for NCCIT cells after 48 h. In summary, treatment of NVP2, SY0351 and YKL-5-124 strongly induced apoptosis in 2102EP and NCCIT cell lines, while TCam2, FS1 and MPAF cells were less sensitive to treatment.

### 3.4. The Molecular Response Is Cell-Line-Specific for NVP2, SY0351 and THZ531 but Not for YKL-5-124 Treatment

In order to determine the molecular mechanisms, we performed RNA-Seq analyses on 2102EP, TCam2 and MPAF cells 1 h and 24 h after treatment (Appendix A). First, we computed the number of commonly deregulated genes for the different cell lines separately for the inhibitors and time points (Figure 4). Interestingly, 2102EP, TCam2 and MPAF exposed to NVP2 1/24 h, SY0351 1/24 h, THZ 1/24 h and YKL-5-124 1 h showed only 0 to 17 commonly deregulated genes (CDG), suggesting that the cell types respond individually to drug treatment. This absence of common mRNA regulation was contrasted by 99 CDGs after 24 h of YKL-5-124 treatment. Analyzing these CDGs in detail revealed upregulation of 8 histone genes. Histone mRNAs are usually not polyadenylated but instead end in a 3′stem loop. The high upregulation thus most likely represents defective 3′-end processing, resulting in the usage of cryptic polyA sites upon CDK7 inhibition [58], suggesting a false positive enrichment of histone mRNAs in mRNA seq analysis. At the same time, expression for several genes (Figure 5A) was commonly downregulated, forming a strong STRING interaction cluster (Figure 5B), which mainly indicates downregulation in mRNA processing and mRNA splicing (Figure 5C). Interestingly, transcripts of immediate early genes such as *EGR1* or transcription factor 1 subunit *JUN* were found to be deregulated in TCam2 cells after 1 h of YKL-5-124 treatment (Appendix A). Further, downregulated gene expression of *ASPH*, *EGR*, *JUN*, *POLR2E*, *CTDP1*, *LY6E* and *PLAUR* was detected. Based on these genes, STRING interaction analysis revealed a highly significant network (Appendix A), which is, according to GO analysis, responsible for decreased transcription activity of RNA polymerase II (Appendix A). MPAF cells exposed to YKL-5-124 for 24 h displayed several downregulated genes related to cell cycle progression (*E2F1*, *E2F2*, *CCNA*, *CCNE2*, etc.), suggesting cell cycle arrest after more than 24 h of treatment (Appendix A).

To further characterize the molecular mechanism of YKL-5-124, we analyzed changes in serine and threonine kinase activity using a peptide microarray. After 1 h of treatment with 2102EP cells, the analysis revealed the death-associated protein kinase 3 (DAPK3) activity to be upregulated (Appendix A), supporting the above-mentioned high potential of YKL-5-124 to induce apoptosis in 2102EP cells. In TCam2 cells, only mild changes in serine/threonine kinase activity have been found globally after 1 h, confirming the diminished effect on the seminoma cell line.

The effects observed after SY0351, NVP2 and THZ531 treatment varied between the cell lines. For SY0351 (CDK7 inhibition), a response to the drug appeared after only 1 h of treatment. In 2102EP cells, functional analysis of STRING interaction networks based on mRNA sequencing data revealed upregulation of the ubiquitin conjugation pathway (*RNF7*, *ASB6*, *UBE2E1*, *NAE1*), which suggests an increase in proteasomal degradation (Figure 6A). Further, expression of genes correlated to stress response (*ERCC1*, *PSMC4*, *GLI1*, *TRIM28*, *FEN1*, *CASP2*, *ENDOG*, *FBXO18*, *WNT5B*, *MAP3K14*) and apoptosis induction was upregulated (*CASP2*, *ENDOG*, *MAP3K14*). In TCam2 cells, after 1 h of SY0351 treatment, STRING interaction analysis revealed initial upregulation of apoptosis and downregulation of translation factor activity (Figure 6B), indicated by the decreased expression of apoptosis-inhibiting factors (*XIAP*, *CFLAR*) and eukaryotic translation initiation factors (*EIF3C*, *EIF2S1*, *EIF5A*), respectively. Mild effects in MPAF cells were observed only after 24 h SY0351 treatment (Figure 6C), implying a weak impact of the drug on the control cell line at the molecular level.

Next, we had a closer look at the effects of CDK9 inhibition caused by NVP2 treatment. After 1 h of NVP2 exposure, two STRING interaction networks suggesting cell cycle arrest and RNA-polymerase-II-specific transcription factor binding were found (Figure 7A). Upregulation of *CDKN1A* expression contributes to the observed cell accumulation in the G1 phase of the cell cycle. At the same time, downregulation of *EGR1*, *IRF1*, *FOSB*, *JUNB*, *SOX2*, *JUN* and *DDIT3* mRNA levels was found, indicating inhibited RNA-polymerase-II-specific DNA-binding transcription activator activity.

In 2102EP cells, treatment for 24 h revealed genes of methyl-CpG-binding domain protein 3-like (*MBD3L2*, *MBD3L3*, *MBD3L5*), TRIM protein (*TRIM43*, *48*, *49*, *49C*) and PRAME (*PRAMEF8*, *9*, *11*, *12*, *14*, *26*) family to be upregulated. This suggests increased DNA-methylation-dependent heterochromatin assembly, ubiquitination activity and negative regulation of transcription, respectively (Figure 7B).

NVP2 treatment in TCam2 cells after 1 h slightly deregulated the apoptotic process and RNA-polymerase-II induced transcription in response to stress due to downregulation of *DDIT3*, *EGR1*, *JUN*, *CBX4*, *NFKBIA*, *DUSP4* and *DUSP6* expression (Figure 7C). STRING analysis of TCam2 cells treated for 24 h revealed a highly significant network consisting of genes representing downregulated mRNA transcript levels. All members (*LSM3*, *SRSF7*, *SNRPB*, *HNRNPC*, *HNRNPH1*, *ALYREF*, *MAGOHB*, *SRSF6*, *HNRNPA1*, *SF3B1*) contributed to RNA splicing or RNA processing, which was obviously downregulated by CDK9 inhibition (Figure 7D).

Peptide microarray analysis revealed a high activity of stress-related and pro-apoptotic kinases (p38α, JNKs) in 2102EP cells 1 h after treatment, which is in accordance with the high rate of apoptosis induced by NVP2 (Appendix A). Interestingly, CDK15 activity was increased, which is in line with the identified upregulation of its corresponding cyclin CCNY. After 24 h, phosphorylation activity of DAPK2 and PIM1/2/3 was increased, indicating apoptosis induction and antagonization of p21 activity, respectively.

Analyzing data from THZ531 treatment revealed upregulation of *PAN2*, *BTG2*, *PABC1* and *PARN* expression responsible for RNA degradation and poly(A)-specific ribonuclease activity (Figure 8A). Treatment for 1 h of TCam2 cells showed downregulation of protein ubiquitination and proteasomal activity (Figure 8B) indicated by decreased expression of 26S-proteasomal subunits (*PSMD7*, *PSMD13*, *PSME4*) and further proteasome-associated transcripts (*RAD23A*, *UBA1*, *NACC1*). The peptide microarray analysis (Appendix A) showed rather increased growth signaling (ERK1/2), cell cycle progression (CDK6) and DNA repair induction (CK1 delta) in 2102EP cells treated with THZ531 for 1 h. Further, 24 h of treatment revealed globally decreased kinase activity, indicating a complete change in cellular response. Kinases displaying top downregulated activity were protein kinase A, which is important for energy metabolism as well as p70S6K and AKT1/2, both part of the PI3K-AKT pathway. In TCam2 cells, the opposite effect compared to 2102EP cells was found. CDK2, ERK1/2 and additionally CDK9 activity was downregulated after 1 h of THZ531 exposure. Similarities were observed at the 24 h time point regarding a global serine/threonine kinase activity downregulation. Interestingly, activity-of-stress-induced protein p38 and several CDKs (CDK9, CDK5, CDK11, CDK1) were found to be decreased in TCam2 cells (24 h) compared to the DMSO control.

Taken together, we could show that a variety of CDK inhibitors effectively reduced viability, induced apoptosis and deregulated cell cycle progression in TGCT cell lines while the fibroblast control cell line was affected only to a limited extend. Investigation of the molecular mechanism of NVP2, SY0351 and THZ531 by RNA Seq analysis revealed cell-line-specific effects. In 2102EP, TCam2 and MPAF CDGs were detected after 24 h of YKL-5-124 treatment, indicating an identical response.

## 4. Discussion

Inhibition of transcriptional CDKs might be a novel promising therapeutic option for patients with TGCTs. In our study, we analyzed cytotoxic effects of seven CDK inhibitors, NVP2, SY0351, YKL-5-124, THZ1, THZ531, dinaciclib and flavopiridol, and one degrader, THAL-SNS-032, on cisplatin-sensitive (2102EP, NCCIT, TCam2) and cisplatin-resistant (2102EP-R, NCCIT-R) TGCT cell lines representing seminomas (TCam2) and embryonal carcinomas (2102EP, NCCIT). Application of the CDK inhibitors/degrader revealed a strong decrease in viability for cisplatin-sensitive and cisplatin-resistant cell lines. 2102EP and TCam2 cells treated with YKL-5-124, SY0351 and NVP2 showed a disturbed cell cycle and apoptosis induction, indicating a potent effect of the inhibitors towards the TGCT cell lines. THZ531 caused cell cycle deregulation only in the 2102EP cells and weak apoptosis induction in 2102EP and TCam2 cells. Analysis of molecular mechanisms demonstrated cell-line-specific responses for NVP2, SY0351 and THZ531 treatment and a general response to YKL-5-124 treatment. We believe that such cell-line (and therefore tumor-entity)-specific responses open up new possibilities of precise treatment of TGCT sub-entities independent of cisplatin resistance.

Interestingly, the Sertoli (FS1) and the fibroblast (MPAF) control cell lines showed strong to moderate and low sensitivity towards the applied compounds, respectively. Taking this into consideration, the CDK inhibitors/degrader might have small to severe side effects on the healthy testis tissue. However, the results suggest that there will be only very small to no effects on fibroblast cells. It has been shown that application of the CDK7/12/13 inhibitor THZ1 led to impaired spermiogenesis in mice after application to testes [59]. However, there is nothing known about effects of NVP2, SY0351, YKL-5-124, THZ531, dinaciclib, flavopiridol and THAL-SNS-032 on testes. Further studies are necessary to evaluate the side effects in vivo.

To our surprise, the molecular response to THZ531, SY0351 and NVP2 treatment varied between 2102EP, TCam2 and MPAF cell lines (Figure 9). For example, THZ531 induced downregulation of the ubiquitin pathway and proteasomal degradation after 1 h of drug exposure of TCam2 cells and increased poly(A) RNA degradation after 24 h of treatment in 2102EP cells. Other studies showed downregulation of DNA damage response gene expression regulated by control of intronic polyadenylation [19,20,27]. Although deregulated poly(A) processing was found in 2102EP cells after 24 h of THZ531 treatment, we did not observe the characteristic downregulation of DNA damage response genes shown to occur in other tumor entities (hepatocellular, ovarian, prostate, breast cancer) [60]. This suggests that TGCT cell lines respond differently compared to somatic cancer types, which might explain the low efficacy of THZ531 in apoptosis induction. In general, germ cells (and their tumors) do not initiate DNA repair but undergo apoptosis instead [61].

In 2102EP cells, the CDK7/12/13 inhibitor SY0351 induced ubiquitin pathway upregulation and prompted stress response after 1 h of treatment, confirming the strong impact on cell viability observed in the XTT assays. Further, inhibition of CDK7 resulted in a lack of CDK1/2/4 phosphorylation, leading to G1- or G2/M-phase cell cycle arrest [31]. Indeed, 2102EP, NCCIT and TCam2 showed a decrease in cells in the S phase and accumulation in the G1 or G2/M phases. For MPAF cells, only weak apoptosis induction was observed after 24 and 48 h of SY0351 treatment.

YKL-5-124 treatment (1 h) caused downregulation of RNA-Pol-II-mediated transcription and immediate early gene transcription in TCam2 cells. Olson et al. found that YKL-5-124 has only a very limited effect on RNA Pol II phosphorylation and activity [31]. Since this effect was detected only after 1 h of treatment, we hypothesize that this finding represents a primary effect which is subsequently compensated by CDK9/12/13. A redundant role in CTD phosphorylation of RNA Pol II of CDK9/12/13 and CDK7 has been shown [31].

GO analysis of RNA sequencing data from 2102EP, TCam2 and MPAF cells after 24 h of treatment with the CDK7-specific inhibitor YKL-5-124 revealed a similar reaction, i.e., the general downregulation of RNA splicing. CDK7 can be regarded as a master regulator of transcription, by virtue of being part of the general transcription factor TFIIH and as part of CAK activating CDKs 9/12/13 by phosphorylation [58]. Rimel et al. observed splicing deregulation induced by treatment with SY0351 (50 nM) in the human leukemia cell line HL60 and proposed CDK7 activity to be crucial for splicing [30]. In our study, 2102EP, TCam2 and MPAF cells treated with YKL-5-124 (100 nM) and TCam2 cells exposed to NVP2 (10 nM) for 24 h but not cell lines treated with SY0351 (10 nM) revealed a disturbed splicing machinery (Figure 9). Of note, YKL-5-124 inhibits CDK7 activity, preventing phosphorylation of CDK9, while NVP2 acts as an ATP-competitive CDK9 inhibitor. These findings suggest, that amongst other important splicing substrates which are phosphorylated by CAK, CDK7 activity affects splicing via CDK9 phosphorylation. STRING analysis of deregulated genes after 24 h NVP2 treatment in TCam2 cells showed a highly compact and significant network comprising proteins such as splicing factor SF3B1, SRSF6, SRSF7, etc., involved in mRNA splicing, mRNA processing and mRNA transport. The importance of CTD phosphorylated RNA Pol II in the splicing process has already been described [62,63]. Further, it has been shown that CDK9 is crucial for full-length transcription, including poly-adenylation and interaction of splicing and export factors with SF3B1, highlighting the role of CDK9 for accurate splicing and mRNA export [64].

Surprisingly, YKL-5-124 elicited moderate to strong apoptosis in the seminoma cell line TCam2 and in EC cell lines (2102EP, NCCIT), respectively. In other studies, it has been shown that CDK7 inhibition alone leads to cell cycle arrest but does not induce apoptosis [31,32]. According to our data, the MPAF control cell line remained almost unaffected. Cell cycle arrest might be initiated after 24 h, indicated by downregulation of several genes associated with cell cycle progression. These findings make YKL-5-124 a promising candidate for targeted treatment of ECs, which seem to exhibit a higher sensitivity towards CDK inhibition compared to other tumor entities and even more importantly towards fibroblasts. In contrast, CDK7/12/13 inhibitor SY0351 revealed a strong apoptosis response in 2102EP, NCCIT and TCam2 cells at a 10-times lower concentration, resulting in a broader response. Most studies describe CDK7-inhibitor-induced apoptosis only in conjunction with another synergistically acting agent, such as 5-fluorouracil, nutlin-3 or a CDK12/13 inhibitor [65,66]. Of note, two selective CDK7 inhibitors, SY-5609 and CT7001, (ICEC9042) are currently investigated in clinical studies. Both inhibitors are tested as single agents and in combination with standard therapy in different tumor types [67,68,69]. In light of our data, we believe it worthwhile to test the drugs on TGCT as well.

The role of CDK9 activity was investigated by CDK9 inhibition (NVP2) and CDK9 degradation (THAL-SNS-032). Although NVP2 and THAL-SNS-032 are characterized by a different mode of action, both compounds revealed highly decreased viability in TGCT cell lines, underlining the relevance of CDK9 for cellular survival. NVP2 treatment (1 h, 10 nM) seemed to induce a prompt stress response of 2102EP and TCam2 cells indicated by upregulation of the cell cycle inhibitor p21 and downregulation of RNA-Pol-II-mediated transcription and immediate early gene transcription (JUN, JUNB, FOSB, EGR1). Lower RNA-Pol-II-mediated expression of immediate early genes is an expected finding after CDK9 inhibition, which is the catalytic subunit of P-TEFb necessary for transcription elongation [70]. P21 mediates the G1- and G2/M-phase cell cycle arrest [71]. In our study, we found accumulation in the G1 as well as G2/M phase in 2102EP and TCam2 cells, respectively, after 20 h of NVP2 treatment. For leukemia cells exposed to 250 nM NVP2 for 6 h, fast apoptosis induction was observed [33]. In TGCT lines, moderate (TCam2) to strong (2102EP) apoptosis was detected, i.e., non-seminoma cells were more affected than seminoma cells. Of note, MPAF cells seemed to be mostly resistant since they displayed only weak apoptosis after 24/48 h of NVP2 treatment. On a molecular level, the peptide microarray analysis assay revealed the activity of apoptosis modulators JNK1/2/3 was highly upregulated after only 1 h NVP2 treatment, indicating apoptosis induction [72].

After 24 h of NVP2 treatment, 2102EP cells showed upregulation of PRAME, TRIM and MBD3L family members. MBD3L2/3/5 are responsible for introducing epigenetic changes by DNA-methylation-dependent heterochromatin formation. PRAME family members in combination with MBD3L2/3 and 5 might induce negative regulation of transcription. TRIM proteins are considered to be involved in modulation of ubiquitin protein ligase activity, indicating upregulated proteasomal degradation. However, it remains elusive how the NVP2-mediated inhibition of CDK9, resulting in alteration of TRIM, PRAME and MBD3L2, 3, 5 levels, contributes to apoptosis induction.

Interestingly, 1 h of NVP2 treatment of TCam2 and 2102EP cells led to upregulation of CDK15 (PFTAIRE2) phosphorylation activity. RNA sequencing analysis revealed significantly increased corresponding cyclin Y expression in 2102EP cells. Park et al. suggested that CDK15 acts as an antagonist of TNF-related apoptosis-inducing ligands [73]. Further, 24 h of NVP2 treatment in 2102EP cells revealed upregulation of oncogenic Pim1/2/3 kinase activity, which is known to be critical for cell survival, proliferation and migration [74]. We speculate that these findings point to a protective mechanism of the cell to prevent apoptosis induction and cell cycle arrest, which fails in the end, as shown by apoptosis after 24 and 48 h.

To sum up the effects caused by NVP2 treatment, we observed cell cycle deregulation and strong apoptosis for TCam2 and 2102EP cells induced after downregulation of RNA splicing and deregulation of chromatin organization, respectively.

## 5. Conclusions

In this study, we investigated the impact and molecular effects of different CDK inhibitors on TGCTs. SY0351 and NVP2 appear to be candidates for a broad application in TGCT treatment since we observed high potency of both compounds towards 2102EP (EC) and TCam2 (seminoma) cell lines. YKL-5-124 turned out to be a highly selective treatment option because the strongest effects were observed for 2102EP cells. Importantly, SY0351, NVP2 and YKL-5-124 displayed a similar cytotoxic effect in wild-type and cisplatin-resistant cell lines, supporting an application which is independent of cisplatin resistance. Unique responsiveness of different cell lines representing different tumor entities shows varying effects on cellular and molecular levels todifferent CDK inhibitors. Thus, exact determination and characterization of tumor composition (seminoma, EC, Cc, YST, Ter, cisplatin-R) and molecular features are crucial for personalized, precise and effective therapy. All in all, CDK inhibitors show a great potential for an alternative and individualized treatment strategy for TGCTs.

## Figures and Tables

**Figure 1 cancers-14-01690-f001:**
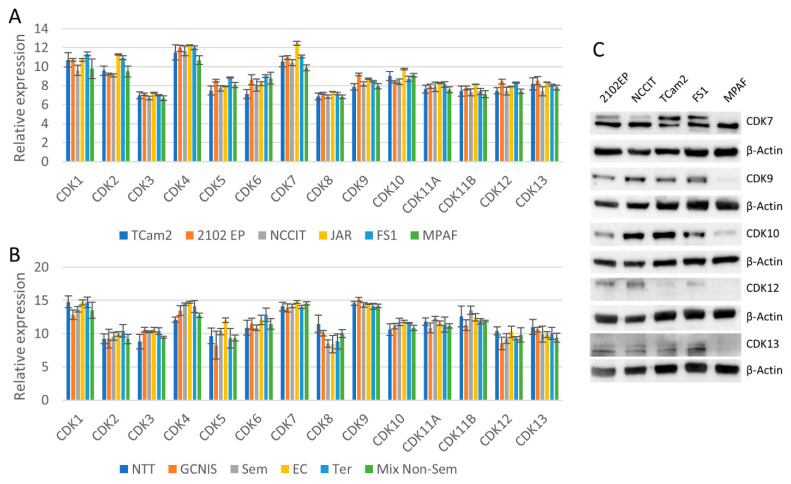
CDK expression in TGCT cell lines and TGCT tissues. (**A**) Meta-analysis of Illumina microarray data from TGCT cell lines (TCam2, 2102EP, NCCIT, JAR), a Sertoli cell line (FS1) and a fibroblast cell line (MPAF). (**B**) Meta-analysis of Affymetrix microarray data from tissues (NTT: normal testis tissues, GCNIS: germ cell neoplasia in situ, Sem: seminomas, EC: embryonal carcinomas, Ter: teratomas, Mix Non-Sem: mixed non-seminomas). (**C**) CDK protein levels in TGCT cell lines and control cells with corresponding β-Actin loading control below each target protein. Original Western Blots can be found at Appendix A.

**Figure 2 cancers-14-01690-f002:**
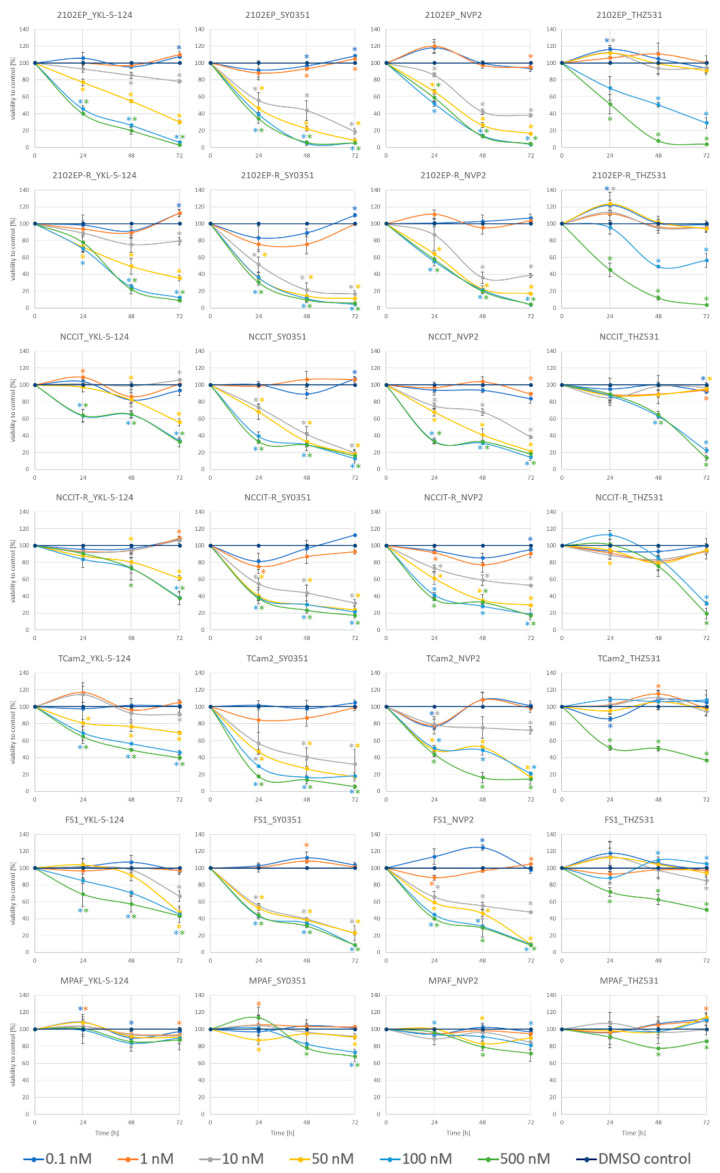
Viability of wild-type and cisplatin-resistant TGCT cell lines after treatment with CDK inhibitors (YKL-5-124, SY0351, NVP2, THZ531). Measurement of viability by XTT assay after single application of CDK inhibitor at 24, 48 and 72 h. DMSO control was applied in all indicated concentrations and referred to as 100% viability. Asterisks indicate significant change in viability between treated and control cells (* *p* < 0.05). Asterisks color code indicate corresponding CDK concentration. *n* = 3–7.

**Figure 3 cancers-14-01690-f003:**
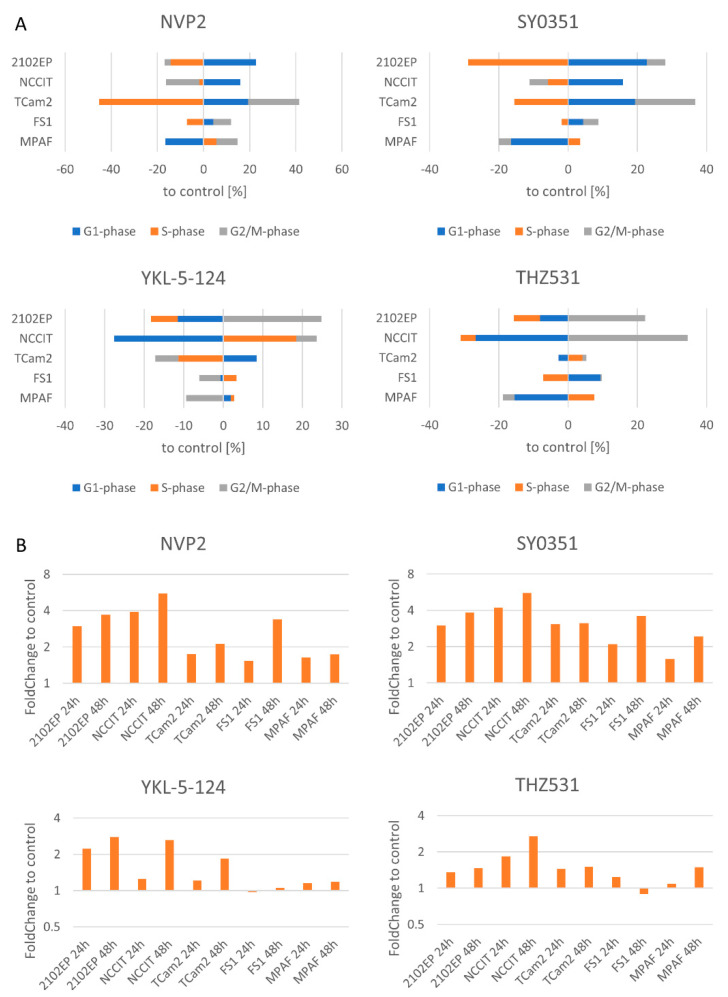
CDK inhibition affects the cell cycle and induces apoptosis. (**A**) Hoechst-FACS-based cell cycle analysis was performed after treatment of the cells with CDK inhibitors (NVP2 10 nM, SY0351 10 nM, YKL-5-124 100 nM and THZ531 100 nM) for 20 h. The distribution of cells in the cell cycle phases is depicted as percent to DMSO-treated control. (**B**) 7AAD/AnnexinV FACS apoptosis analysis after 24 h and 48 h of treatment (same concentrations as indicated above).

**Figure 4 cancers-14-01690-f004:**
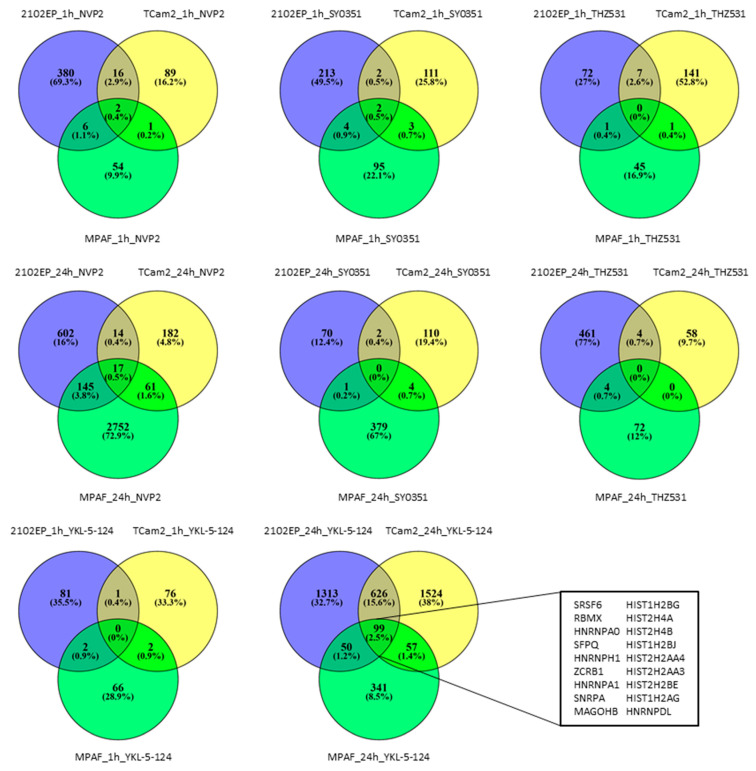
mRNA sequencing analysis revealed commonly deregulated genes in 2102EP, TCam2 and MPAF cell lines only after YKL-5-124 treatment. Significantly differential expressed genes were compared between the different cell lines individually for NVP2, SY0351, YKL-5-124 and THZ531. Selected commonly differential expressed genes are indicated for YKL-5-124 treatment (24 h). For all samples, *n* = 3.

**Figure 5 cancers-14-01690-f005:**
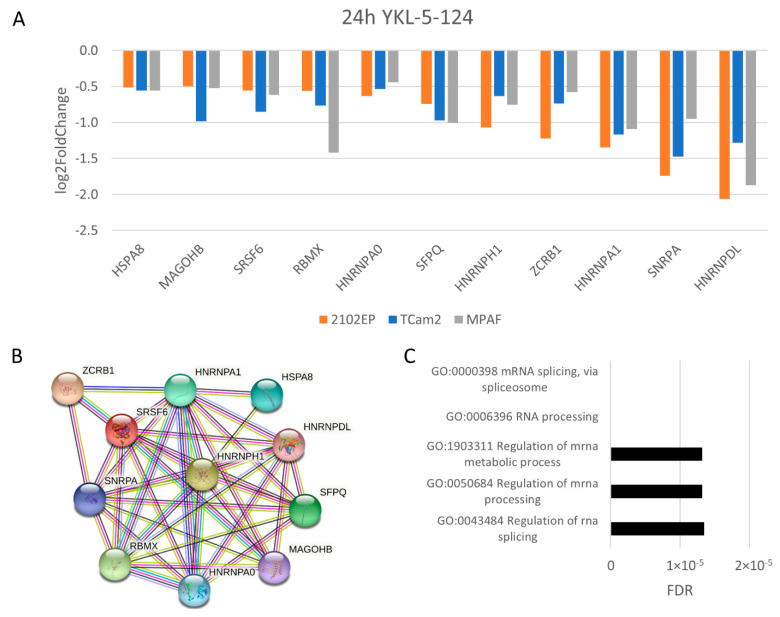
Elucidating molecular effects of YKL-5-124 treatment. (**A**) Commonly downregulated genes after 24 h of treatment (100 nM). (**B**) STRING-based interaction analysis and (**C**) Gene Ontology analysis of commonly downregulated genes. For all samples, *n* = 3.

**Figure 6 cancers-14-01690-f006:**
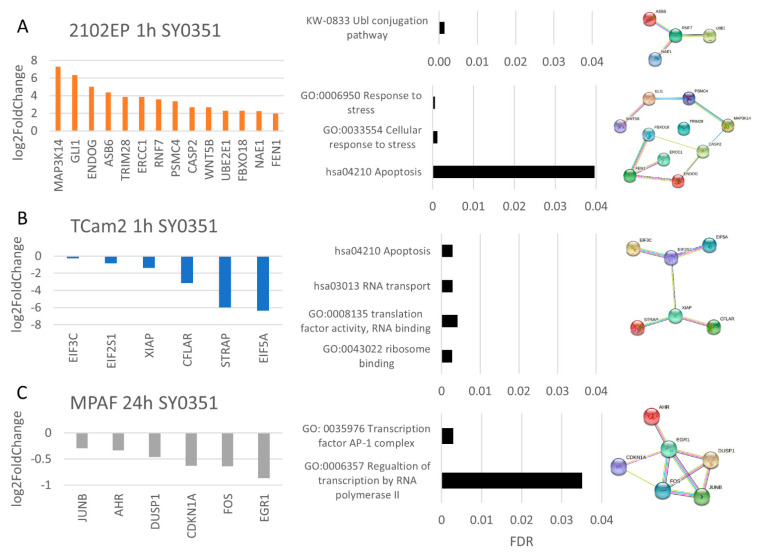
SY0351 treatment reveals cell-line-specific response. RNA-seq data after SY0351 treatment (10 nM) shown as differential expression, GO and STRING analysis for (**A**) 2102 EP (1 h), (**B**) TCam2 (1 h) and (**C**) MPAF (24 h). For all samples, *n* = 3.

**Figure 7 cancers-14-01690-f007:**
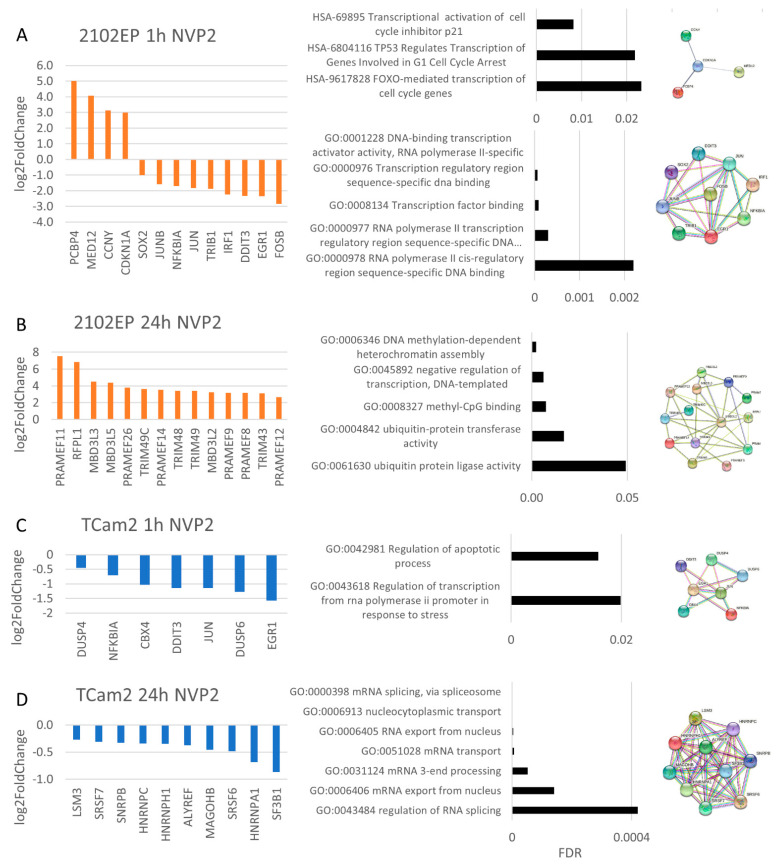
NVP2 treatment indicated cell-line-specific effects. RNA-Seq evaluation was performed via Reactome pathway analysis/Gene Ontology analysis and STRING interaction analysis in (**A**) 2102EP cells after 1 h of treatment, (**B**) 2102EP cells after 24 h, (**C**) TCam2 cells after 1 h and (**D**) TCam2 cells after 24 h. For all samples, *n* = 3.

**Figure 8 cancers-14-01690-f008:**
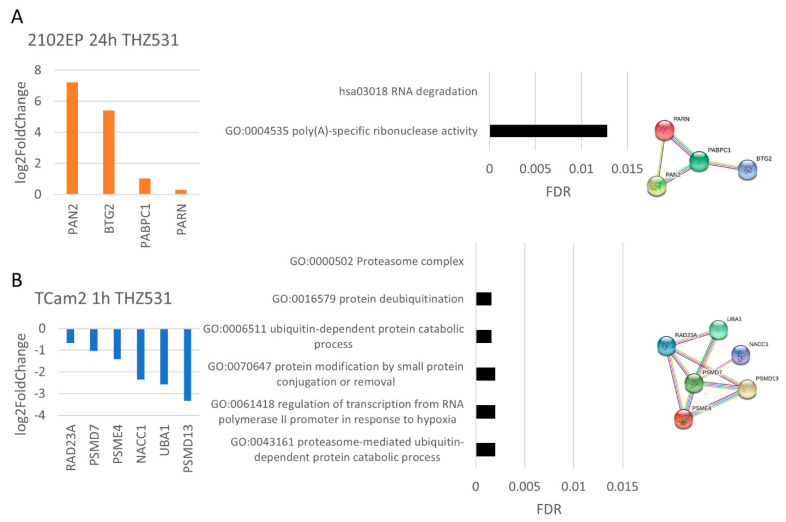
THZ531 treatment induced cell-line-specific responses. Differential gene expression data evaluated via Reactome pathway/Gene Ontology analysis and STRING interaction analysis are presented. Treatment of (**A**) 2102EP cells for 24 h and (**B**) TCam2 cell for 1 h (100 nM THZ531). For all samples, *n* = 3.

**Figure 9 cancers-14-01690-f009:**
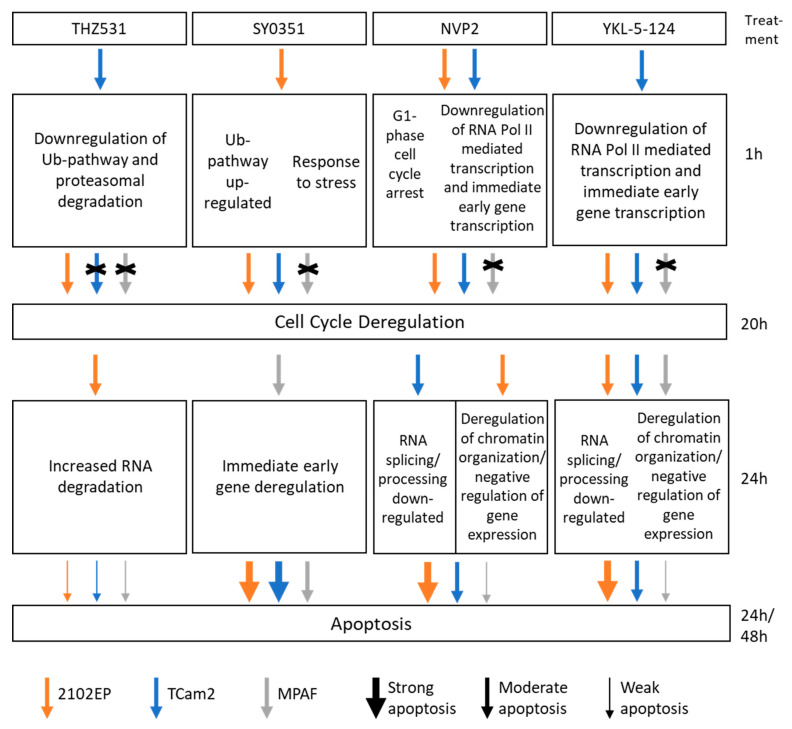
Schematic summary of effects in TGCT and fibroblast cell lines induced by CDK inhibitor treatment. Treatment with SY0351, NVP2 and YKL-5-124 results in cell cycle deregulation and apoptosis in TGCT cells while exposure to THZ531 is less toxic to all cell lines.

**Table 1 cancers-14-01690-t001:** Primary and secondary antibodies used in this study.

Target	Company	Species	Dilution	Order No.
CDK7	Invitrogen, Waltham, MA, USA	Rabbit	1:1000	PA5-34791
CDK9	Cell signaling, Danvers, MA, USA	Rabbit	1:1000	2316T
CDK10	Cell signaling, Danvers, MA, USA	Rabbit	1:500	36106S
CDK12	Merck, Darmstadt, Germany	Rabbit	1:500	ABE1861
CDK13	Antibodies-online.com (01/2022)	Rabbit	1:1000	ABIN6130965
β-Actin	Merck, Darmstadt Germany	Mouse	1:35,000	a5441
Anti-mouse HRP	Agilent Technologies (Dako), Santa Clara, CA, USA	Rabbit	1:750	P0260
Anti-rabbit HRP	Agilent Technologies (Dako), Santa Clara, CA, USA	Goat	1:2000	P0447

**Table 2 cancers-14-01690-t002:** IC50 values of TGCT and control cell lines. The calculation was performed based on the XTT viability. The values were subgrouped according to very high potency (green), high potency (yellow), moderate potency (red) and low potency (white).

	IC50 [nM]							
Cell Line and Treatment Time	NVP2	SY0351	YKL-5-124	THZ531	THZ1	Dinaciclib	Flavo-Piridol	THAL-SNS-032
2102EP_24 h	502.7	38.7	317.6	>1000	95.6	76.8	>1000	164.15
2102EP_48 h	10.5	7.5	43.6	179.6	26.3	3.4	39.94	39.51
2102EP_72 h	6.1	6.7	17.2	74.5	6.7	0.8	18.01	34.82
2102EP-R_24 h	884.1	15.9	>1000	>1000	235.3	102.2	>1000	
2102EP-R_48 h	11.2	3.4	30.5	227.5	16.3	2.0	21.85	
2102EP-R_72 h	8.6	6.4	23.6	163.9	9.9	1.1	7.56	
NCCIT_24 h	90.6	104.5	>1000	>1000	>1000	593.5	710.47	149.39
NCCIT_48 h	40.7	19.1	>1000	>1000	92.9	21.8	21.85	50.62
NCCIT_72 h	6.5	9.0	121.1	96.6	9.3	1.5	34.07	30.09
NCCIT-R_24 h	103.7	24.9	>1000	>1000	929.4	314.8	638.83	
NCCIT-R_48 h	17.1	14.0	>1000	>1000	145.0	19.6	106.83	
NCCIT-R_72 h	12.2	11.6	178.1	137.2	16.2	4.3	43.28	
TCam2_24 h	215.7	21.6	>1000	>1000	>1000	35.1	>1000	139.83
TCam2_48 h	60.0	8.9	>1000	>1000	>1000	11.5	581.81	76.70
TCam2_72 h	16.1	8.6	263.4	>1000	847.0	1.6	272.74	39.57
FS1_24 h	98.2	85.4	>1000	>1000	>1000	392.6	>1000	>1000
FS1_48 h	33.5	30.1	>1000	>1000	>1000	85.7	695.75	>1000
FS1_72 h	8.9	7.7	95.2	>1000	>1000	14.3	91.47	>1000
MPAF_24 h	>1000	>1000	>1000	>1000	>1000	>1000	>1000	>1000
MPAF_48 h	>1000	>1000	>1000	>1000	>1000	>1000	>1000	>1000
MPAF_72 h	>1000	>1000	>1000	>1000	>1000	>1000	>1000	>1000

## Data Availability

Raw and processed mRNA sequencing data generated in this study have been deposited in the Gene Expression Omnibus database under accession code (GSE198248).

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
