# Peer review of "Transcriptional CDK Inhibitors as Potential Treatment Option for Testicular Germ Cell Tumors"

_cancers, 2022, doi:10.3390/cancers14071690_

Round 1
Reviewer 1 Report
Thank you very much for giving me this opportunity to review this article. The followings are my comments.
-The ordinate axes of Figure 1A and 1B start from 6. These may give readers visual misunderstandings; therefore, starting points of ordinate axes should be zero.
-Some references in the main text are written as "Error! Reference source not found."
-There are two "Figure 2" in this manuscript. These misdescriptions should be correctable by pre-submitting proofreading. Did the authors proofread their manuscript before submitting it?
Author Response
lease see the attachment

Reviewer 2 Report
Add inhibitor to keyword “trascriptional CDK…”
Please better describe the main charactheristics of type II TGCT and the relationship among type II TGCT, seminomas and non seminomas
What is the meaning of “*” in figure 1?
Check line 247
In each panel of figure 2, please Add the curve of untreated cells over the time and the statiscal significance. In addition, to improve the readers understanding could be showed in figure 2, only the drugs abile to affect cell viability and the others in supplementary materials.
Perform statistical analysis in all experimental and add the method used.
add statistical significance
Avoid the use of color and discriminate the several treatment in an other way.
Prior to analyze cell cyclette by FACS, did you performed cell sincronization?
Round 2
Reviewer 2 Report
Authors improved the manuscript
according to my suggestions